# Spectral Flow Cytometry: The Current State and Future of the Technology

**DOI:** 10.3390/ijms26125911

**Published:** 2025-06-19

**Authors:** E. A. Astakhova, A. S. Gubaeva, D. A. Naumova, A. E. Egorova, A. A. Maznina, I. G. Rybkina, I. M. Osmanov, D. V. Tabakov, O. N. Mityaeva, P. Yu. Volchkov

**Affiliations:** 1Federal Research Center for Original and Prospective Biomedical and Pharmaceutical Technologies, Baltiyskaya Street 8, 125315 Moscow, Russia; 2Moscow Center for Advanced Studies, Kulakova Street 20, 123592 Moscow, Russia; 3Morozovskaya Children′s City Clinical Hospital, 4 Dobryninskiy pereulok 1/9, 119049 Moscow, Russia; 4Bashliaeva Children’s Municipal Clinical Hospital of the Moscow City Health Department, Geroev Panfilovtsev Street 4, 125373 Moscow, Russia; 5Department of Fundamental Medicine, Lomonosov Moscow State University, Lomonosovsky Prospekt 27, 119992 Moscow, Russia; 6Moscow Clinical Scientific Center N.A. A.S. Loginov, Entuziastov 86, 111123 Moscow, Russia

**Keywords:** spectral flow cytometry, spectral unmixing, deep immunophenotyping, fluorescence

## Abstract

Flow cytometry is a powerful and widely used tool for the analysis of various cell populations, but its capabilities are severely limited by the need to apply correction of fluorescent signals from near or similar fluorochromes when analyzing multicolor panels. Spectral flow cytometry extends the capabilities of classical cytometry by reading the full fluorescence spectrum of fluorophores and their subsequent spectral separation. This significantly increases the number of markers analyzed in a single panel and thus allows for more in-depth studies of cell populations. In the age of big data analysis, this represents a serious advantage of spectral cytometry and can significantly increase its use in scientific and clinical practice. This review describes the principle of spectral cytometry, advantages and limitations of the method, and summarizes the newest deep immunophenotyping panels developed and validated for spectral cytometry.

## 1. Introduction

Currently, flow cytometry is a widely used method in clinical and scientific practice. This method is based on measuring multiple parameters from cell suspension or particles as they flow by one in a fluid stream through a laser beam. This method enables the study of various cells of eukaryotic organisms of animal and plant origin, cultured cell lines, microbial populations, and others. The distinctive feature of flow cytometry is its high productivity (measurement of more than 10,000 cells per second) presenting the possibility of studying the phenotype and genotype of cells, as well as the processes occurring in them [1,2].

Flow cytometry has been actively developed for over 50 years. During this time, the technology has evolved from the measurement of a single parameter to a multiparameter analytical tool that allows a large amount of information to be obtained about the samples being tested [3]. The most commonly used instruments in scientific practice are those with the ability to analyze up to 20 markers [4], but the most advanced have the ability to analyze more than 20 colors. In fact, about 10 parameters are usually analyzed simultaneously on classical flow cytometers. While 10–20 years ago this number was considered sufficient, today, in the age of big data, this opinion is questioned [5].

The rapid development of flow cytometry is inextricably linked to the development of immunology and the study of the role of the immune system in the pathogenesis of many human diseases, especially cancer and autoimmune diseases, since in these diseases, the impaired functioning of the immune system plays one of the main roles [6,7]. One of the key events in the field of therapy of cancer and autoimmune diseases was the discovery of immune checkpoints—molecules on the surface of immunocompetent cells that can regulate the direction and strength of the immune response. The use of new drugs based on monoclonal antibodies, aimed at blocking or, conversely, at activating these molecules, significantly expands the possibilities of immunotherapy [8]. At the same time, there is a need to select biomarkers that can be used to assess and predict the efficacy of immunotherapy. Often, this choice can be made after a complete analysis of the functional status of immune cell populations and observation of its dynamics in the process of drug administration. The depth of immunophenotyping required for these tasks can only be achieved by increasing the number of parameters read from cell populations.

At present, profiling of the immune system based on single-cell mRNA sequencing (ssRNA-seq, single-cell RNA-sequencing) is becoming widespread, providing information on the expression of all genes of a cell. However, mRNA levels correlate poorly with the expression of proteins, which are traditionally used to determine the types of immune cells, which may lead to an incorrect assessment of the functional activity of a particular cell [9,10]. In this regard, there is a need to confirm ssRNA-seq data using laboratory methods, such as cytometry.

Currently, new platforms for multiplex analysis of cells using expressed surface and intracellular markers are being actively developed: CITE-Seq, mass cytometry (CyTOF), and spectral cytometry. CITE-Seq allows for the analysis of markers through the use of oligonucleotide-labeled antibodies [11]. Mass cytometry is a combination of flow cytometry with mass detection of the label mass using a mass spectrometer. This method uses antibodies labeled not with fluorophores, as in classical cytometry, but with metals. These are mainly isotopes of heavy metals and rare-earth metals (lanthanide isotopes), i.e., those that are not normally present in biological objects. Serious limitations of mass cytometry are low throughput, high cost, limited reagents and equipment availability, and the impossibility of cell sorting [12]. Finally, spectral cytometry is a method that is the closest to classical cytometry. In its modern form, the capabilities of spectral flow cytometry were demonstrated by Paul Robinson in 2004 [13]. The development of spectral cytometry as a method is described in more detail in the review by I.A. Vorobiev [14]. The peculiarity of the technology lies in reading the full spectrum of fluorophores, which significantly increases the resolution of the method relative to classical flow cytometry [15]. This review discusses the differences between spectral and classical flow cytometry, as well as the capabilities of spectral cytometry.

## 2. Design of the Classical and Spectral Flow Cytometer

Flow cytometers consist of three principal parts: the fluidics, the optical system, and the electronics [16]. Laser combinations in conventional and spectral cytometry do not have significant differences; the most spread are 3- and 5-laser combinations. Customization for laser wavelength is now available in limited conventional cytometers (e.g., Symphony™ systems). W. Telford notices that violet laser (as an example) has greater value for spectral system compared to conventional cytometry if the panels do not have violet-excited fluorochromes. In a spectral system, a violet laser will excite all fluorochromes, not just the ones with an excitation maximum in the violet range. The resulting fluorescent signal will be used for enhanced unmixing [17].

The fundamental differences between classical and spectral flow cytometers are in the detection system. Classical cytometers are equipped with optical filters (dichroic mirrors and bandpass filters) that separate and direct the light emitted by fluorophores to the appropriate detector. Each detector (typically a photomultiplier tube, PMT) is tuned to a narrow band of wavelengths that approximates the emission peak of the known fluorophores. In this way, the “one detector–one fluorophore” approach is realized. The bandwidth of most band-pass filters ranges from 20 to 50 nm. PMTs detect photons at the corresponding wavelength and narrowing this range can introduce errors due to the stochastic nature of the photoelectric effect, which becomes more significant at the lower detection limit. Thereby, the fluorescent signal (350–850 nm) can be divided to a limited number of detectors (typically 10–12) [18]. In addition, an increase in the number of detectable parameters is required, a serious complication of the optical system of filters and detectors. Thus, on average, a flow cytometer registering signals from 12 fluorophores contains 12–14 independent detectors and more than 40 optical filters [19]. A more complex detection system significantly increases the cost of the device [20].

In modern advanced cytometers, these issues are partially mitigated by employing more sensitive detectors (high-performance PMTs, avalanche photodiodes (APDs)), an increased number of spatially separated lasers, improved quality of optical filters, and establishment of precise laser delay. As a result, according to manufacturers, modern conventional flow cytometers can analyze more than 20 parameters in their maximal configuration, such as the BD FACSymphony™ A5 Cell Analyzer (50 detectors, 30 fluorescent and 20 optional), the Bio-Rad’s ZE5 (30 detectors), and the BC CytoFLEX LX (21 detectors).

Spectral cytometers collect the entire emission spectrum of each fluorophore over a wide range of wavelengths (Table 1). This is achieved by using a prism or diffraction grating to scatter the emitted light, which is then captured by an array of highly sensitive detectors (on average 40, Table 1) [15]. Thus, spectral flow cytometers are less optically complex because there is no need to install a complex filter system configuration. This makes spectral flow cytometers cheaper to manufacture [21].

## 3. New Fluorescent Molecules

Detection of the full fluorescence spectrum and application of spectral unmixing of overlapping spectra enables the expansion of the panel of fluorophores up to 50 or more parameters. It becomes possible to use fluorophores with overlapping spectra, provided that their full spectral profiles are distinguishable. However, the implementation of this possibility is difficult due to the limited number of commercially available fluorescently labeled antibodies and spectrally different dyes [4]. Almost all fluorescent dyes are suitable for spectral cytometry—fluorescent proteins, small organic, quantum dots, polymeric, tandem dyes, and others. Development of new fluorochromes is actively evolving, as evidenced by the appearance of many new dyes on the market over the last 5–10 years. We summarized the main dye groups, focusing on the most modern dyes (Table 2). However, reviewing all available new dyes is nearly impossible [22].

### 3.1. Fluorescent Proteins (FPs)

The first group of FPs includes proteins such as GFP, YFP, and mCherry, often used in reporter systems. Another group of dyes based on protein molecules are phycobiliproteins: phycoerythrin (PE), alophycocyanin (APC), and peridinin-chlorophyll protein complex (PerCP). They are characterized by their exceptional brightness, which continues to make them a popular choice for researchers.

All fluorescent proteins are also characterized by shortcomings due to their protein nature (sensitivity to solvent, temperature, pH). Protein dyes are also prone to photobleaching and have a limited shelf life [23]. The main advantage of using fluorescent proteins, as well as small organic molecules as dyes, is the wide choice of labeled antibodies on the market [24].

### 3.2. Tandem Dyes

This type of dye utilizes two fluorophores that are a FRET pair. A fluorescent protein or organic polymer can act as the fluorescence donor and a small organic fluorophore can act as the acceptor.

Tandem dyes have a broad Stokes shift up to 300 nm and can access the far-red spectral region, as demonstrated by BioLegend′s APC/Fire™ 810 and PE/Fire™ 810. The primary disadvantages of tandem dyes include their light sensitivity, instability, and susceptibility to degradation under various conditions [23]. Although the manufacturers claim improved photo- and thermal stability and brightness of new tandem dyes [25,26], issues with degradation or unexpected spectrum signatures remain problematic [27].

In addition, tandem dyes exhibit batch-to-batch and inter-manufacturer variability, which makes it necessary to record reference spectra for each conjugate, use appropriate reference spectra for each measurement, and update references more frequently than with non-tandem dyes. Figure 1 demonstrates differences in spectral unmixing quality when using “relevant” reference spectrum and reference spectrum from the same tandem dye but conjugated with a different antibody.

### 3.3. Small Organic Molecules

Small organic molecules can be found in fluorescin, coumarin-based dye families, Alexa Fluor™, BODIPY™, Vio™, and many others. “The color palette” of low molecular weight organic dyes covers almost the entire spectrum from UV to visible and near-infrared. They are often used for conjugation to antibodies and are convenient for intracellular labeling due to their small size, but are generally characterized by low brightness.

Manufacturers continue to improve organic fluorochromes and introduce new product lines. For example, the Spark line of dyes (Biolegend) was developed considering the spectral profiles of other dyes and therefore provide information able “to fill the gaps” between the emission peaks of existing fluorochromes.

### 3.4. Quantum Dots (Qdots, QD)

QD are nanometer semiconductor particles with a special structure [28]. Their emission region depends on the particle size and spans the wavelength range from 525 to 800 nm. The appearance of these dyes on the market resulted in a significant increase in the number of simultaneously used colors [29]. They are bright and stable dyes, have a broad Stokes shift and narrow, symmetrical emission peaks. Their unique spectrum is also an advantage for spectral cytometry. The disadvantage of these dyes is their broad excitation spectrum, partial loss of properties during long storage, and high cost. Quantum dots are only suitable for surface labeling and the number of commercially available antibody conjugates is limited.

### 3.5. Polymer Dyes

Fluorochromes comprising a π-conjugated polymer chain capable of absorbing and emitting light in specific regions of the spectrum belong to this type of markers. An important advantage of these markers is their extremely high extinction coefficients due to the cooperative behavior of multiple optical subunits along the polymer chain.

The advent of the first line of Brilliant Violet polymer dyes raised polychromatic cytometry to a new level by making an 18-color palette available to researchers [30]. Brilliant Ultraviolet also managed to unlock the potential of the UV laser by offering fluorochromes excited in the UV region. When several polymer dyes are used together, the manufacturer recommends using a special buffer to avoid nonspecific polymer–polymer interactions.

### 3.6. Multimeric DYES

Many of the strategies for creating new dyes are based on the multimerization of fluorophores, which can significantly increase the brightness of the tag. The techniques for creating such fluorescent tags are discussed in detail in a review [31].

Different lines of dyes, in which the principle of multimerization is implemented, have been proposed in recent years. For instance, the Miltenyi company introduced the VioBright line [32], in which fluorophores are fixed on a polyester framework.

### 3.7. DNA-Based Dyes

DNA-based dyes are a stable and compact structure that spontaneously assembles to from single-stranded DNA molecules and includes several fluorophores that form FRET pairs [33]. The appeal of using DNA as a backbone for fluorophores is attributed to the properties of the molecule itself and the excellent understanding of these properties [34]. The DNA molecule is stable, which provides stability during staining and degradation during fixation. The predictable structural characteristics of the molecule allow the creation of dyes with a unique fluorescence spectrum by adjusting the distance between fluorophores. By adding more fluorophores, it becomes possible to adjust the brightness of the dyes. In addition, attaching a precise number of fluorophores opens up the possibility of quantifying antigens on the cell surface [35]. DNA-based dyes are potentially biocompatible and minimally toxic.

DNA-based fluorochromes cannot be used in the presence of DNA intercalating dyes, as their binding leads to quenching or alteration of the fluorescent signal [36]. In addition, such dyes may bind to immune cells, so the use of special blocking buffers supplied by the manufacturer is essential [37].

### 3.8. Polymer Dots

An alternative to fluorochromes based on linear polymers are polymer dots (PDots). Like quantum dots, they are semiconductor nanoparticles but have a completely different structure [38]. They cover a wide range of colors, outperform the polymer dyes BV, BUV, Super Bright, and traditional fluorophores in brightness, and have narrow excitation and emission peaks, a unique type of spectrum. In addition, fluorochromes of this family are characterized by stability, are resistant to the action of fixing agents, and do not require special buffers.

### 3.9. Fluorescent Tags of the Future

Fluorescent carbon materials, including graphene and carbon quantum dots, have received a great deal of attention in recent years. They are of particular interest due to their tunable spectral properties. They are water-soluble, biocompatible, chemically inert, and characterized by high photostability [39]. These materials have great potential for the development of new fluorescent markers.

### 3.10. Features of Using Dyes in Spectral Flow Cytometry

The features of using dyes in spectral cytometry are predominantly related to using relevant reference spectra of dyes that can be changed under various conditions. Firstly, they may unpredictably vary by batch or manufacturer, as shown in Figure 1. Secondly, some fluorochromes show different emission spectra observed on cells or beads [40]. For example, cells, unlike calibration beads, make a false positive contribution to the YG4-561 channel [24], so it is important to experimentally validate the beads or cells used as single-stained controls to achieve optimal unmixing of a fully stained sample. Finally, it is also worth mentioning that fixation/permeabilization reagents can alter fluorochrome spectra, as well as other aspects of staining. For best results, antibody titration should be performed using the same buffer, as for fully stained samples.

## 4. The Spectral Cytometry Pipeline

General guidelines for high-quality cytometry data—including instrument setup, sample preparation, antibody titration, panel design, etc. [16,41,42]—are also suitable for spectral cytometry. A key distinction in the spectral cytometry workflow is the requirement to record reference emission spectra for every conjugated antibody and fluorescent marker used in the panel (Figure 2).

The autofluorescence (AF) spectrum of an unstained sample should also be recorded. AF arises from natural fluorophores in cells, such as NAD(P)H, FAD, collagen, elastin, etc. [43]. AF spectrum is later extracted from the total fluorescent signal during multicolor staining analysis. The major problem with AF that is more crucial in spectral cytometry rather than conventional, is its variability which depends on cell type, their metabolic state, of senescence stage, injury status, sample preparation, etc. [43]. It means that the most representative AF spectrum must be used to distinguish true fluorescence signals from background autofluorescence. Spectral cytometers’ software can automatically extract just one AF spectrum per sample that is insufficient for heterogeneous samples such as tissues in which different cell types may have different AF spectra. For instance, unanticipated autofluorescence (AF) spectra in tissue—when not properly extracted from the fluorescent signal—may overlap with fluorochromes, affecting the labeling of rare but critical markers.

Post-acquisition approaches to address this issue are now being actively developed [44,45]. These approaches aimed, in general, to identify pure cell subset-specific AF spectra within mixed signals using dimensional reduction and incorporate them into the unmixing algorithm. If the cell types in suspension are not well-characterized or predictable, unbiased clustering should be used to reveal all unique subsets—not just those selected manually.

To summarize, the spectral cytometry pipeline includes unique features such as reference spectrum acquisition, spectral unmixing, and, when required, advanced AF extraction. Other steps, including multicolor staining and data analysis remain similar to conventional cytometry.

## 5. Deconvolution of Fluorescence Signal: Compensation and Spectral Unmixing

As discussed above, in conventional flow cytometry, the principle “one detector–one fluorophore” is realized. However, detectors designed to register the signal from one fluorophore usually receive photons from other fluorophores whose fluorescence spectra overlap [46]. “Spillover” of fluorescence beyond one′s detector leads to signal mixing. To separate signals in classical flow cytometry, a mathematical algorithm based on construction of “spillover” and “compensation” matrices is applied. The rows (*i*) in the spillover matrix *Mij* correspond to detectors and columns (*j*) correspond to fluorochromes being detected. Each cell of this matrix represents spillover coefficient (how much signal from fluorochrome *j* appears in detector *i*). This data is obtained from single-stained controls. A compensation matrix is used to correct for the spillover and is the inverse of the spillover matrix. Both of these matrices are square because number of detectors and fluorofores are equal [47]. The goal of compensation is to transform the data such that the values from a single detector correspond to only one fluorochrome.

As the number of fluorophores in a single panel increases, compensation becomes more and more difficult and is often manually adjusted. There is a wide variety of recommendations for compensation optimization, with their number increasing every year [48]. For example, using beads partially mitigates problems with compensation but the quality of these beads should be previously tested (its AF, the brightness of signal that they produce). Sometimes, beads-based correction leads to loss/gain populations when applied to cells [49].

Spectral cytometry provides data from a greater number of detectors than fluorescent markers used in the panel collecting the full continuous emission spectrum of used fluorochromes. To perform unmixing calculations, the method of Ordinary Least Square is the most widely used [50]. The calculation estimates the linear contribution (matrix C) of each fluorochrome′s reference spectrum (matrix A) to the mixed spectrum from multicolor staining (matrix Y) using the equation Y = A × C. These matrices correspond to signals collected by the detectors. Y matrix consists of 1 column. Matrix A has multiple columns, each corresponding to a fluorochrome′s reference spectrum used in the panel. Unlike spillover and compensation matrices (which are square), these matrices are rectangular because there are more detectors than fluorochromes. In other words, the goal of spectral unmixing is to determine the contribution of each fluorochrome to the mixed spectrum.

## 6. Clustering, Visualization, and Analysis of Multicolor Cytometry Data

In classical cytometry, 2D diagrams are traditionally used for data representation, which are constructed by sequential gating of cell populations [51]. Spectral cytometry can be used to determine a much larger number of parameters than classical cytometry, so traditional 2D diagrams are insufficient to reflect the multidimensional relationships between cell populations [24].

The most popular algorithms for reducing the dimensionality of data and representing them in the plane are PCA, t-SNE, and UMAP. PCA finds the orthogonal components of the feature space along which the data has the highest variance, which are then used to represent in a low dimensional space [52]. It is supposed that linear methods such as PCA are mostly unsuitable for visualizing cytometric data because they cannot accurately represent nonlinear dependencies [53].

t-SNE and UMAP, unlike PCA, are nonlinear methods that seek to preserve the local neighborhood of points in multidimensional. UMAP is computationally more efficient than t-SNE and better preserves the global cluster structure [54]. The application of the t-SNE algorithm to cytometry data and especially the search for rare populations is not optimal because the algorithm performs poorly with a large number of observations (events), which is common in cytometry [53]. Therefore, new dimensionality reduction mechanisms based on t-SNE are being developed for specifically cytometric data, such as viSNE [49] and opt-SNE [53].

A recent review [55] discussed two large categories of clustering tools used for cytometric data analysis: unsupervised (ACCENSE, DensVM, SPADE, FlowSOM, PhenoGraph, and others) and supervised (openCyto, Flowlearn and others). Such tools organize cells into clusters based on the same expression of their markers. Unlike manual gating, which is performed based on the researcher′s knowledge of the order of the procedure, these clustering tools work “blindly”, which enables the detection of novel and rare phenotypes in a mixed population of cells. Supervised tools, unlike unsupervised tools, can not only cluster cells, but also annotate the resulting clusters.

## 7. Deep Immunophenotyping by Spectral Cytometry

Spectral cytometry is becoming an increasingly popular tool for simultaneous detection and analysis of a large number of protein markers at the level of single cells. Spectral cytometry′s potential as a method is explored extensively in the field of in-depth phenotyping of immune cells [56]. Improved definition of immune cell subpopulations and their characteristics enables better understanding of the immune system’s role in disease pathogenesis and therapy of autoimmune [57], infectious diseases [58], and cancer immunotherapy [59]. Particularly important is the ability to determine many cellular parameters for difficult to access, quantitatively limited samples (biopsy material, bone marrow aspirate, pediatric peripheral blood).

One of the challenges of modern immunology is the identification and validation of immunologic biomarkers for routine clinical use that correlate with the progression of cancer or autoimmune disease, with a positive response to immunotherapy or vice versa, associated with drug resistance [60]. It has been shown that the expression levels of single markers (e.g., PD-1 for nivolumab therapy) can act as such biomarkers [61] or several checkpoint molecules at once (e.g., LAG-3 and PD-1), which is expressed in the need to assess the expression of targeting molecules on different populations of the immune system to predict the effect on the antitumor response [62].

The degree of activation and functional activity of immune cells is also a biomarker. For instance, staining of T cells for cytokines, chemokines, and degranulation markers in response to antigen-specific stimulation is widely used [63]. Changes in immune cell populations are another type of biomarker [64]. So, in cancer, tumor immune infiltration and shifts in immune cell ratios (e.g., naive cells, Tregs, MDSCs, effector cells, exhausted cells) help predict and assess immunotherapy efficacy [65].

The acquisition of comprehensive information on the expression of a wide range of markers by a particular cell enables a more precise attribution of the cell to one or another population. For instance, a significant biomarker is the number and activity of regulatory T cells (Treg) and their subpopulations in human tumors and in the peripheral blood of cancer patients, as well as in autoimmune diseases [66,67]. However, using only FOXP3 is not enough to define this cell population, as Miyara et al. have shown functionally distinct Treg subpopulations (resting Treg cells—CD45RA^+^FoxP3^low^; activated Treg cells—CD45RA^−^FOXP3^high^; nonsuppressive T cells—CD45RA^−^FOXP3^low^) [68]. Moreover, CD8^+^FOXP3^+^ cells have been described [69]. A combination of at least seven markers—CD3, CD4, CD25, CD127, FOXP3 with Ki67, and CD45RA—have been recommended to clarify Treg status [56]. In addition, evaluation of additional markers such as CD39, CTLA-4, GITR, ICOS, LAP, GARP, PD-1, PD-L1, and several others also play an important role [70,71,72,73]. Each combination of these markers may indicate a different functional activity of the Treg subpopulation and indicate changes that have occurred during the course of disease progression.

Another striking example is minor B-lymphocyte populations (immature/transitional B cells) (~2.5% in blood) in plasma cells (~2% in blood) [74], which require the use of more than 10 markers for accurate identification [75]. The difficulty of phenotyping the MDSC population, which plays a role in tumor escape from the immune response and suppression of the hyperactivation of the immune response in inflammatory processes, is the subject of many studies [76]. In addition, a huge number of different minor subpopulations that play roles in disease pathogenesis and progression have been described and can be used as biomarkers in oncology and immunology [77,78].

In recent years, deep immunophenotyping has been applied to hematological malignancies. Currently, MRD detection and the identification of malignant lymphoid cells by flow cytometry rely on 8- to 13-color panels, which can distinguish surface markers of undifferentiated cells, such as CD34, CD33, and the negative expression of CD38 and CD45RA, among others [79,80]. Spectral cytometry has enabled the expansion of these panels, with two multiplex systems—20-color and 24-color—being developed [81,82]. Validation against standard 8- to 10-plex panels showed very high correlations between the main defined populations and MRD status. As the authors note, minor discrepancies were primarily due to the number of events acquired, debris removal, and doublet exclusion. Despite these challenges, the inclusion of additional markers (e.g., CD117, CD133, CD49f, and CD90) allows for finer discrimination of differentiation stages and detection of changes in bone marrow composition. This approach holds promise for investigational use and as a diagnostic tool in MDS, which represents a pre-leukemic phase [83].

Another promising application of multiparameter spectral cytometry is assessing the biological function and phenotypic relevance of CAR-T cells. Cadinanos-Garai et al. developed a 36-marker spectral flow cytometry panel for integrated profiling of CAR-T cell production. Their study revealed functional differences between Day 5 and Day 10 of T-cell expansion, showing comparable cytotoxicity but distinct activation and checkpoint profiles. A key finding was that cryopreservation of the CAR-T cell product only modestly affected stem cell memory, activation, and metabolic markers, while preserving the overall phenotype and cytotoxic function. Beyond direct assessment of viability and functional activity, this approach could help fine-tune therapy—such as through direct injection of an optimized cell pool—maximizing efficacy based on individual immunotype [84].

Other appealing applications of multiparameter cytometry are the use of peptide-MHC multimers to identify T cells of a given specificity, or to screen for multiple specificities. For instance, mass cytometry has been used to show that slow progression of type 1 diabetes, as opposed to rapid progression, is associated with an “exhausted” phenotype of specific CD8+ T cells [85]. Spectral cytometry has the capability to detect and characterize such rare populations.

## 8. The Newest Multicolor Panels for Spectral Cytometry

The issue of standardization in flow cytometry has improved significantly in recent years, but the development of standard protocols for immunophenotyping remains extremely important [86]. In the last few years, Optimized Multicolor Immunofluorescence Panels (OMIP) have been developed for the phenotyping of peripheral mononuclear cells, T cells, and monocytes using spectral flow cytometry. Examples of the use of these and other multicolor panels have been presented in a review [14]. Here we summarize recently published (2023–2025) multicolor panels developed for spectral flow cytometry.

It is worth noting that some modern advanced conventional cytometers (discussed above) have more than 20 detectors and can potentially analyze a number of parameters comparable to those listed in Table 3. However, the published 20+ color panels optimized for conventional cytometers are in the minority compared to those for spectral systems. Most high-parameter conventional panels have been validated for BD FACSymphony A5: 25-color panel for human intestinal tissue [87], 27-color panel for PBMC phenotyping [88] and major leukocyte populations in fixed whole blood [89], 28-color for moDCs in mice spleen [90], etc. A cross-platform comparison by Heubeck et al. evaluated the conventional BD FACSymphony A5 and spectral Cytek Aurora using 25/26-color PBMC panels [91]. Authors conclude that both platforms have similar effectiveness, with population frequencies showing exceptionally high correlation (≥0.989). In the spectral data, some markers exhibited slightly better population separation, but this did not affect gating strategies or frequency outcomes.

## 9. Limitations and Challenges of Spectral Cytometry

Like any method, spectral cytometry has a number of limitations and difficulties in application. The development of large multicolor panels is still a complex process and requires considerable knowledge and technical skill. It remains impossible to completely eliminate errors in spectral separation, so basic rules of panel design must be followed for successful experiments, regardless of which instruments are used.

Theoretical calculations’ similarity indexes of dye spectra frequently fail to predict actual separation quality, particularly for tandem dyes. The limited commercial availability of antibodies labeled with diverse fluorochromes also restricts the full potential of spectral cytometry. Spectral cytometry requires recording a reference spectrum of each used fluorochrome that is inherent in conventional cytometry. Single-color control spectra are then stored in the software library and can be used repeatedly in one series of experiments, reducing pre-measurement preparation time. However, reference spectra must be updated regularly to account for potential fluorochrome degradation over time [101] and between experiments. It consumes additional reagents and increases workload. The challenge in analyzing data obtained with a spectral cytometer is associated with the use of software for visualizing multidimensional data only ex post facto. In addition, these methods are sensitive to minor changes in the original datasets. As discussed above, autofluorescence is one more source of difficulty in the interpretation of spectral cytometry data.

## 10. The Future of Spectral Flow Cytometry

Spectral cytometry is already accepted as a revolutionary technology in the field of analyzing a large number of cellular parameters [5]. However, the success and mass application of this technology depends on the development of several aspects.

Unlike in CyTOF, in spectral cytometry, cells remain whole after analysis, so there is a possibility to sort them. The major challenge for implementing sorting is performing spectral unmixing in real time, which requires a number of hardware and software modifications [24]. The success of cell sorting based on spectral analysis technology will determine its place in biological research. For instance, it is expected that with spectral sorters, researchers will be able to validate their transcriptomic data—comparing RNA and protein data at the level of individual cells. The currently available spectral sorters on the market are summarized in Table 4.

There are expectations for the development of improved, high-precision computational tools for analyzing spectral cytometry data, the introduction of cluster visualization into the software that is provided by the instrument manufacturer, and advanced tools for AF extraction, as mentioned above.

Since conventional cytometry has already been approved for clinical use, spectral cytometry is just beginning to be used in clinical laboratories in China and some European countries (e.g., Cytek Biosciences) [102]. Obviously, gaining regulatory approval for spectral cytometry in clinical practice will broaden its application and provide valuable data.

In solving these problems, spectral cytometry has great potential to become an integral part of the analysis of cell populations for solving diverse problems in the field of immunology.

## 11. Conclusions

The primary advantage of spectral flow cytometry is the ability to simultaneously analyze more parameters in a single sample than in classical cytometry. This is achieved by reading the full fluorescence spectrum of fluorophores. Many researchers currently consider spectral cytometry excessive for routine laboratory practice. However, its spectral unmixing capability significantly reduces both compensation setup time and operator training requirements. At present, the application of spectral cytometry is limited by the number of fluorescence-labeled antibodies available on the market and issues with AF extraction, but these fields are actively developing. The application of spectral cytometry for deep immunoprofiling in cancer, autoimmune, and infectious diseases will allow the identification of new biomarkers for therapy of these diseases.

## Figures and Tables

**Figure 1 ijms-26-05911-f001:**
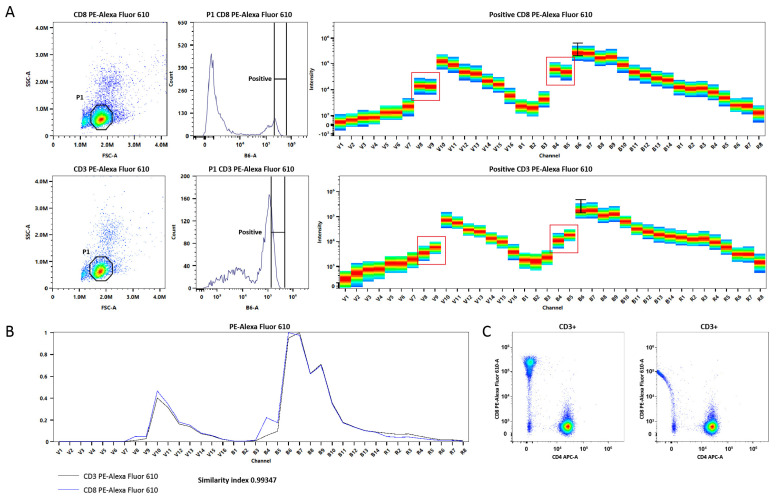
Troubles in spectral unmixing of tandem dyes. (**A**) Reference spectra of CD8- and CD3-PE Alexa Fluor 610. Red squares highlight differences in the same fluorochrome spectra conjugated to different mAbs. (**B**) Spectral profiles of PE-Alexa Fluor 610 tandem dyes. (**C**) Representative unmixed cytograms. PBMCs were stained with CD45-PerCP, CD3-PE, CD4-APC, CD8-PE-Alexa Fluor 610. **Left** panel: unmixing was performed using the correct reference spectrum (from CD8-PE-Alexa Fluor 610). **Right** panel: unmixing was performed using an incorrect reference spectrum (from CD3-PE-Alexa Fluor 610), demonstrating the need for additional compensation. Colors of the spectra indicate intensity of signal (blue is low-intensity, red is high-intensity signal).

**Figure 2 ijms-26-05911-f002:**
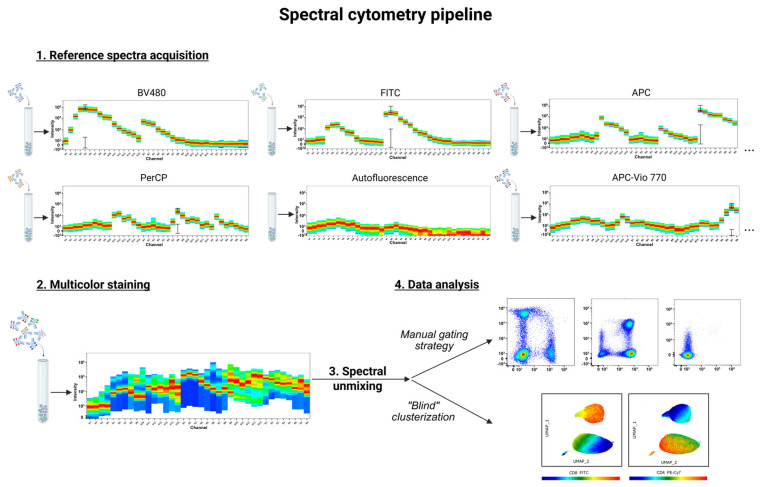
Key steps in spectral cytometry: reference spectra acquisition, multicolor staining, spectral unmixing (optional compensation), and data analysis. Colors of the spectra indicate intensity of signal (blue is low-intensity, red is high-intensity signal).

**Table 1 ijms-26-05911-t001:** Selected spectral cytometers available on the market and their characteristics.

Characteristics	Sony SA3800	Sony ID7000	Cytek Northern Lights	Cytek Aurora	Agilent NovoCyte Opteon	Attune^TM^ Xenith^TM^	BD FACSymphony™ A5 SE
**Number of lasers and their wavelengths**	up to 4–405/488/561/638 nm	up to 7–320/355/405/488/561/637/808 nm	3– 405/488/640 nm	5–355/405/488/561/640 nm	up to 5–349/405/488/561/ 637 nm	6–349/405/488/561/ 637/781 nm	5–355/405/488/561/637 nm
**Detection system**	Set of 10 prisms, 32-channel PMT array	32-channel PMT arrays, individual PMTs	CMWD *	CMWD *	CMWD *	PMT array	Cascade square PMT array
**Number of detection channels, range of detectable wavelengths**	FSC/SSC + 32F **	FSC/SSC + 184F **	FSC/2 SSC + 38F **	FSC/2 SSC + 64F **	FSC/2 SSC + 73F **	3 FSC/3 SSC + 51F **	FSC/SSC + 48F **
**Number of colors in the panel**	Not specified	44 or more	Up to 24	Up to 40	Up to 45	Up to 32	Up to 40
**Software**	Sony Spectrum Analyzer		SpectroFlo		Agilent NovoExpress	Attune NxT Cytometric Software	BD FACSDiva

*—Matrix of semiconductor detectors. **—Fluorescent (channels).

**Table 2 ijms-26-05911-t002:** Selected new fluorescent dyes.

Type	Fluorochromes	Manufacturer
Small organic molecules	Spark™, Spark PLUS™	Biolegend
	Vio™	Miltenyi
	eFluor™ 450, eFluor™ 660	Thermofisher
Tandem dyes (fluorescence donor—protein)	Fire™	Biolegend
	PerCP-eFluor 710, APC-eFluor 780	Thermofisher
	Astral Leap™	Biotium
Tandem dyes (fluorescence donor—polymer)	BD Horizon™ BV605, BV650, BV711, BV786	BD Biosciences
Polymer dyes	SuperBright™	Thermofisher
Multimers	VioBright™	Miltenyi
	RealBlue™, RealYellow™	BD Biosciences
	KIRAVIA™	Biolegend
DNA-based dyes	NovaFluor™	Thermofisher
Polymer dots (PDots)	StarBright™	Bio-Rad

**Table 3 ijms-26-05911-t003:** The newest multicolor panels for spectral cytometry.

Cell Populations	Color Number	Instrument	References
Human PBMC			
B-, T-, γδ-T-, NKT-like, NK-cells, monocytes, basophils, DCs, ILCs	45	5-laser (355, 405, 488, 561, 640 nm) Cytek Aurora	[92]
CD4 Treg: naive, memory, TR1-like, activated cells. CD4: TFH, TEMRA, TCM, TEM, naive, TSCM, Th1, Th2. CD8: TEMRA, TCM TEM, naive, TSCM	31	5-laser (355, 405, 488, 561, 640 nm) Cytek Aurora	[93]
T-, B-, NK-cells, DCs, ILCs	50	7-laser (320, 355, 405, 488, 561, 637, 808 nm) Sony ID7000 and 5-laser (349, 405, 488, 561, 637 nm) BD FACSDiscover S8	[94]
Subpopulations of T-, B-cells, DCs, plasmacytoid DCs, basophils	30	5-laser (355, 405, 488, 561, 640 nm) Cytek Aurora	[95]
27 T-cell subpopulations, 5 B-cell subpopulations, NK-, NKT-cells	27	4-laser (405, 488, 561, 640 nm) Cytek Aurora	[96]
Subpopulations of dendritic cells, NK cells, monocytes, MDSCs	25	4-laser (405, 488, 561, 640 nm) Cytek Aurora	[96]
Fresh lysed whole human blood			
CD4-, CD8-, γδ-T-cells, Treg, NKT-like; B-, NK-cells, DCs, monocytes.	40	Not stated, probably 5-laser (355, 405, 488, 561, 640 nm) Cytek Aurora	[27]
Platelets	16	3-laser (405, 488, 640 nm) Cytek Northern Lights	[97]
Subpopulations of dendritic cells, NK cells, monocytes, MDSCs	35	6-laser (320, 355, 405, 488, 561, and 637 nm) Sony ID7000	[98]
T-cell and B-cell subpopulations	34	6-laser (320, 355, 405, 488, 561, and 637 nm) Sony ID7000	[98]
B-cells	24	BD FACS Canto II, Cytek Northern Lights (laser config is not stated)	[82]
10 T-cell subpopulations, B-cells, progenitor cells, ILCs	32	4-laser (405, 488, 561, and 640 nm) Cytek Aurora	[96]
Mice tissues			
Mouse T-, B-, NK-cells, innate lymphoid and dendritic cells, monocytes, macrophages, basophils, neutrophils, eosinophils	40	5-laser (355, 405, 488, 561, 640 nm) Cytek Aurora	[99]
B-cells, subpopulations of T-cells, pDCs, cDCs, macrophages, neutrophils, monocytes	13	5-laser (355 nm, 405 nm, 488 nm, 561 nm, 637 nm) BD FACSymphony S6, 5- laser (355 nm, 405 nm, 488 nm, 561 nm, 637 nm) Sony ID7000, 5-laser (355, 405, 488, 561, 640 nm) Cytek Aurora	[100]

**Table 4 ijms-26-05911-t004:** Selected spectral sorters available on the market and their characteristics.

Characteristics	Bigfoot™ Spectral Cell Sorter	Cytek Aurora™ CS System	FP7000 Spectral Cell Sorter	BD FACSDiscover™ S8 Cell Sorter
**Manufacturer**	Invitrogen	Cytek Biosciences	Sony	BD Biosciences
**Number of lasers and their wavelengths**	9 (349/405/445/488/532/561/594/640/785 nm)	5 (355/405/488/ 561/640 nm)	Up to 6 (320/349/405/488/561/637 nm)	5 (349/405/488/561/637 nm)
**Number of colors in the panel**	up to 60	up to 40	over 44	up to 38
**Sorting possibilities**	1.5, 5, 15 and 50 mL tubes, 10× chips, plates (96-, 384- and 1536-well), PCR strips	1.5, 5, 15 mL tubes, plates (96- and 384-well)	1.5, 5, 15 and 50 mL tubes, plates (6-384-well)	5 mL tubes, plates (96- and 384-well plates)

## Data Availability

Not applicable.

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
