# Peer review of "Spectral Flow Cytometry: The Current State and Future of the Technology"

_ijms, 2025, doi:10.3390/ijms26125911_

Round 1

Reviewer 1 Report

Comments and Suggestions for Authors

The review article by E.A. Astakhova and Coworkers illustrates the current state-of-the-art and the future developments of spectral flow cytometry (SFC), a diagnostic technology applicable in many clinical settings, where the simultaneous analysis of multiple cellular indicators is necessary.

GENERAL COMMENTS

I don't know if this review article has been specifically invited by the Editor or if it has been spontaneously proffered by the Authors.

The general tone of the present review article seems rather a biased endorsement of SFC, mostly focused on the limitations of conventional flow cytometry (FCM).

The text lacks the necessary, balanced pairwise comparison of the favorable or unfavorable features of either platforms, that can help the reader in fully obtaining a clearer picture of the current technical advantages and disadvantages of the two technologies.

Crucial technical items of SFC are not described or explained in a sufficient and comprehensible detail. Namely, spectral unmixing and its similarity/difference to conventional compensation is just cited without explanation; normalization using calibration beads; the need of establishing the emission spectrum of each conjugated antibody or fluorescent marker. This omission is a major limitation of the present paper.

The instruments reported in Table 1 are unexplainedly limited to two Sony and two Cytek models, whereas the current repertoire of commercially available platforms is wider (i.e. including the Agilent NovoCyte Opteon, the Attune Xenith, the hybrid Becton Dickinson A5 SE and FASCDiscover S8, to name a few). This gap cannot be present in a review article of this type.

The paragraphs describing fluorochromes appear redundant in their present form. More relevant would have been a more detailed analysis of how some representative fluorochromes may behave differently in SFC and in FCM under various conditions. The short paragraph involving PE-Alexa Fluor 610 tandem- conjugated antibodies depicted in Figure 1 is a nice example, but other similar problems might have been described or cited.

Although the majority of papers on SFC-based deep immunophnotyping are focused on immunological studies, the usage of SFC in hematological malignancies should not be missed.

This issue is still in its infancy, but equally amenable of multicolor deep phenotyping. The following papers may be of help to give a more comprehensive view of the current applications in hematological oncology: Matthes T. Int J Mol Sci 2024; 25: 2847.   García-Aguilera G. et al, Cells 2024; 13: 1891.  Cadinanos-Garai A. et al, Molecular Therapy 2025; 33(5): 2291-2309.

SPECIFIC COMMENTS

Abstract: The sentence "...its capabilities are severely limited by the need to apply compensation..." is frankly speaking inappropriate. Compensation is a major issue in conventional FCM, but it is effectively managed by modern beads and software. By the way, disturbing issues similar to compensation are also present in SFC.

Introduction, lines 39-41: The sentence "...due to the difficulties of applying compensation in multicolor panels on classical flow cytometers, about 10 parameters are analyzed simultaneously..." is both incorrect (see above) and obscure. Please reword.

Introduction, lines 73-74: "...advantages and disadvantages of these methods have been previously described in a review". Please indicate the citation of this review.

Introduction. A major issue in the controversial 'competition' between SFC and FCM is the limit above which the simultaneous number of markers to be analyzed may suggest the use of SFC instead of conventional FCM. This important issue has not been clearly included in this review.

The most advanced conventional FCMs can accommodate up to 5 lasers and 18-27 different fluorescence detectors, so the stick is set at a very high level. Please discuss adequately this important aspect.

Chapter 2. Design of the classical and spectral..., line 88: The correct abbreviation of photomultiplier is PMT. Please amend.

Chapter 2. Design of the classical and spectral..., lines 95-96: The sentence "... classical cytometers are often limited to simultaneous measurement of a relatively small number of parameters..." is a risky statement, since 18-27 parameters do not seem a 'relatively small' number, as specified above. Please reword.

Chapter 2. Design of the classical and spectral..., lines 102-103: The sentence "... too many channels can limit the capabilities due to photon statistics..." is obscure. Please reword and clarify.

Chapter 2. Design of the classical and spectral... The technical complexity related to the laser setup has not been taken into account at all in this chapter. This is an issue that is common to both SFC and FCM. Please add an adequate section on this important technical issue.

Table 1. As mentioned, the table should be enriched with all the currently available intruments, not just show these four.

Chapter 2. New fluorescent molecules, is actually Chapter 3. Please amend.

New fluorescent molecules, lines 110-112: The sentence "... the limited number of commercially available fluorescently labeled antibodies and spectrally different dyes..." is another risky statement. The current repertoire of fluorescently labelled antibodies and fluorescent probes is surely in the thousands. Moreover, trying to summarize them in a review article is an almost impossible task. The Authors may better cite the recent volume by C. Ortolani, Flow Cytometry Today, Springer 2023; ISBN978-3-031-10835-8, https://doi.org/10.1007/978-3-031-10836-5, which describes in great detail all the possible fluorescent molecules for SFC and FCM, omitting Table 2 and the paragraphs from 2.1 to 2.8.

Figure 1 should be better put in Chapter 3 (Actually Chapter 4, due to the wrong numbering), with an adequate paragraph explaining in detail the technical problems of autofluorescence elimination, normalization and spectral unmixing, that are the major technical issues in SFC.

Line 240: "...transfusion coefficients" are unknown to me. Please correct.

Chapter 6: The sentence "The lack of standardization in flow cytometry has traditionally hampered its use in multicenter clinical trials..." could have been acceptable 20 years ago. Many efforts have been made in the meantime to standardize and harmonize FCM analyses in many different clinical settings. Please reword.

Table 3, page 11, last block. The color number is missing.

Chapter 7, Limitations. The need of pre-recorded emission spectra of antibody panels should be clearly highlighted. Moreover, the need of recording multiple single-color emission spectra whenever a new experiment with different antibodies is being developed should be stressed, which may cause a heavy workload. These points are not inherent in conventional FCM and should be adequately commented.

Chapter 8, Table 4 (Wrongly labelled as Table 3 at page 11, line 383. Please add 'SORTERS' to the legend, for sake of clarity.

Chapter 8, lines 384-386: The sentence "... to combine different types of analysis not only the phenotyping of cells, but also the determination of their cell cycle stage, metabolic and structural components..." is wrong, because the cited features can be equally accomplished by conventional FCM.  Please omit or reword.

Chapter 8, end. Please cite the fact that SFC instruments are currently not yet CE-IVD or FDA cleared for clinical diagnostic procedures.

Please take care in the correct numbering of chapters and tables.

Reference n.1 is in Russian, but it should be accessible by an international readership. Please replace with another review paper, for instance the ref.#15.

Comments on the Quality of English Language

Some sentences are imprecise or even obscure, and have been specifically highlighted in the comments for Authors.

Author Response

Dear Reviewer #1, 

Thanks for your valuable comments. Our responses are provided in the attached Word file.

Reviewer 2 Report

Comments and Suggestions for Authors

In this manuscript, Astakhova and collaborators explore the potential of spectral cytometry, outlining its key strengths and limitations relative to other methods and traditional cytometry.

The authors cover the most important points. However, I found some weak points in the general information on the method. I recommend adding a brief general description of cytometry, particularly in the introduction, to help non-expert readers understand the method's main features and the differences between traditional and spectral cytometry.

I have summarized the main suggestions below.

- In general, references need to be included and expanded. For example, in R100-103, reference 18 is given for two successive concepts.

-In R75–77 (ref 12 and 13), R186 (ref 29), and R346 (ref 13): The citations are given very schematically. I suggest providing a brief description to expand on them.

Paragraph 5: 'Deep immunophenotyping by spectral cytometry' (R320–325). Minor clarification: The listed markers allow the identification of Tregs and their subsets, as also reported by Miyara and Sakaguchi in 2009 (DOI 10.1016/j.immuni.2009.03.019). I suggest expanding the concept to clarify and better explain this sentence and the next one, 'In addition... an important role'.

The same goes for R327: the sentence is unclear. What is meant by 'minor B lymphocyte populations'? Give examples.

Paragraph 6 is somewhat one-sided. I suggest either expanding the discussion on OMIP panels or integrating the paragraph elsewhere in the manuscript by referring to Table 3.

-R384–R386: This should be explained more clearly, because traditional cytometry also allows both immunophenotyping of cells and the simultaneous analysis of cell cycle, activation, and proliferation.

Minor points:

- Check for minor errors throughout the text.

Please ensure that the tables are correctly numbered. There are two tables named Table 3.

Please pay attention to the formatting in Table 3 (which refers to sorters).

In Table 3 (OMIP) referring to Panel 2, there is some text written in the Cyrillic alphabet. Furthermore, I recommend making all acronyms explicit in the caption.

Author Response

Dear Reviewer #2, 

Thanks for your valuable comments. Our responses are provided in the attached Word file.

Round 2

Reviewer 1 Report

Comments and Suggestions for Authors

The Authors have submitted a revised version of their originally rejected manuscript.

The majority of the remarks have been tackled and corrected, also including some additional valuable references, and the paper is now more balanced and informative.

A few sentences are still to be amended or reworded, as follows:

1) Lines 88-91: The citation of Telford's paper needs rewording, because it is obscure in its present form. Maybe violet has been confused with ultraviolet.

2) Line 114: MoFlo Astrios is not marketed anymore, and can be omitted.

3) Line 278: "...are actively developed last time.." has to be reworded.

The wording and the syntax of the main text may benefit from an accurate review by a mother tongue expert.

Comments on the Quality of English Language

The wording and the syntax of the main text may benefit from an accurate review by a mother tongue expert.

SOme sentences requiring rewording have been indicated.

Author Response

(The authors gave the same response as above.)

Reviewer 2 Report

Comments and Suggestions for Authors

The authors responded to all my comments, and the manuscript was improved.

Author Response

Dear Reviewer #2, 

Thanks for your  positive assessment of our revised manuscript.